# Early exposure to farm dust in an allergic airway inflammation rabbit model: Does it affect bronchial and cough hyperresponsiveness?

Amandine Divaret-Chauveau[1,2,3]*, Laurent Foucaud[2], Bruno Demoulin[2], Cédric Teston[2], Pauline Loison[4], Pierre Le Cann[5], Cyril Schweitzer[2,6], Marcelo De Carvalho Bittencourt[7,8], Frédéric Mauny[3,9], Silvia Demoulin-Alexikova[2,10]

1 Pediatric Allergy Department, University Hospital of Nancy, Vandoeuvre-lès-Nancy, France, 2 EA3450 Développement Adaptation et Handicap (DevAH), University of Lorraine, Vandoeuvre-lès-Nancy, France, 3 UMR 6249 Chrono-environment, CNRS and University of Franche-Comté, Besançon, France, 4 INRS (National Research and Safety Institute for the Prevention of Occupational Accidents and Diseases), Laboratory of Aerosol Metrology, Vandœuvre-lès-Nancy, France, 5 EHESP School of Public Health, Inserm, IRSET (Institut de Recherche en Santé, Environnement et Travail), UMR_S 1085, Université de Rennes, Rennes, France, 6 Department of Pediatric Lung Function Testing, Children's Hospital, University Hospital of Nancy, Vandœuvre-lès-Nancy, France, 7 Immunology Department, University Hospital of Nancy, Vandoeuvre-lès-Nancy, France, 8 Université de Lorraine, UMR 7365 CNRS, IMoPA, Nancy, Vandoeuvre-lès-Nancy, France, 9 Unité de Méthodologie en Recherche Clinique, Epidémiologie et Santé Publique, CIC Inserm 143, University Hospital of Besançon, Besançon, France, 10 CNRS, Inserm, CHU Lille, Institut Pasteur de Lille, U1019-UMR9017-CIIL-Centre d'Infection et d'Immunité de Lille, University of Lille, Lille, France

* a.chauveau@chru-nancy.fr

**Data Availability Statement:** All relevant data are within the paper and its Supporting Information files.

## Abstract

### Introduction

Over the past 50 years, the prevalence of allergic respiratory diseases has been increasing. The Hygiene hypothesis explains this progression by the decrease in the bio-diversity of early microbial exposure. This study aims to evaluate the effect of early-life farm exposure on airway hyperresponsiveness and cough hypersensitivity in an allergic airway inflammation rabbit model.

### Method

A specific environment was applied to pregnant rabbits and their offspring until six weeks after birth. Rabbits were housed in a pathogen-free zone for the control group and a calf barn for the farm group. At the end of the specific environmental exposure, both groups were then housed in a conventional zone and then sensitized to ovalbumin. Ten days after sensitization, the rabbit pups received ovalbumin aerosols to provoke airway inflammation. Sensitization to ovalbumin was assessed by specific IgE assay. Cough sensitivity was assessed by mechanical stimulation of the trachea, and bronchial reactivity was assessed by methacholine challenge. The farm environment was characterized by endotoxin measurement.

**Funding:** The present study was supported by Scientific Council of ARAIRLOR (Association Régionale d'Aide aux Insuffisants Respiratoires de Lorraine). The funders had no role in study design, data collection and analysis, decision to publish, or preparation of the manuscript.

**Competing interests:** A.D-C. reports support from ARAIRLOR for the present manuscript; a contract with the French Public Agency ANSES as an expert in allergy and pediatric; consulting fees for expertise in pediatric allergy for Stallergens, Aimmune Therapeutics and ALK; and support for attending meetings from Mead Johnson, Nutricia, Aimmune Therapeutics and Novartis Pharma SAS. F.M. reports a grant from the French public agency ANSES and participation on a Data Safety Monitoring Board for the clinical "Propila-Rifax". All other authors have no conflict of interest in relation to this work.

## Results

A total of 38 rabbit pups were included (18 in the farm group). Endotoxin levels in the farm environment varied from 30 to 1854 $EU.m^{-3}$. There was no significant difference in specific IgE values to ovalbumin ($p = 0.826$) between the two groups. The mechanical threshold to elicit a cough did not differ between the two groups ($p = 0.492$). There was no difference in the number of cough ($p = 0.270$) or the intensity of ventilatory responses ($p = 0.735$). After adjusting for age and weight, there was no difference in respiratory resistance before and after methacholine challenge.

## Conclusion

Early exposure to the calf barn did not affect cough sensitivity or bronchial reactivity in ovalbumin-sensitized rabbits. These results suggest that not all farm environments protect against asthma and atopy. Continuous exposure to several sources of microbial diversity is probably needed.

## Introduction

Over the past 50 years, the prevalence of allergic respiratory diseases has been constantly increasing in industrialized countries. This progression makes it a major public health problem and the development of prevention methods must today constitute a health priority. The Hygiene hypothesis explains this progression by our modern lifestyles, particularly by the decrease in the biodiversity of microbiological exposure in the perinatal period [1, 2]. The reduction in exposure to microbial agents during pregnancy and the first years of life may favor Th2-mediated allergic disorders [3, 4].

The European birth cohort PASTURE (Protection against Allergy: STUdy in Rural Environments), set up in the early 2000s, has focused on the protective effect of growing up on a dairy farm against Th2-driven allergic responses [5–7]. Up until now, several international cross-sectional studies have demonstrated that early farm exposure protects against allergic diseases, atopic sensitization, and asthma [8, 9]. Early exposure during pregnancy and first years of life seems to be a window of opportunity for immune homeostasis [10]. However, most of the time, farm exposure is continuous during the entire childhood so it is not known if early farm exposure followed by an exposure break is sufficient to provide protection against allergic diseases.

Even though exposure to farm dust, with a high microbial diversity content and a high endotoxin level, is a well-known protective factor from the farm environment [11–13], it is not the only pathway to biodiversity of microbial exposure. Exposures to different animal species during pregnancy, high food diversity in the first year of life, and a diet rich in dairy products, especially cheese, have also been involved in the protective effect of farming lifestyles on allergic diseases and asthma [14–17]. Moreover, the host genetic background in the host-microbiome crosstalk may play an important role [18].

Significant advances in understanding this 'farm effect' have been explored using an approach that compares and contrasts traditional and modern farming environments and populations with closely comparable genetic compositions [19]. However, setting up a clinical therapeutic trial in this direction turned out to be very complex and its attempt failed. Facing the current impossibility of carrying out a clinical therapeutic trial aimed to 'apply' the farm

environment to a cohort of newborns, experimental models would represent a good alternative.

The aim of this study is to set up early-life farm exposure in an allergic airway inflammation rabbit model in order to test the hypothesis that such exposure protects from airway hyperresponsiveness and cough hypersensitivity.

The use of a rabbit model sensitized to ovalbumin in order to model allergic airway inflammation is quite appropriate for this study. In addition to being the model of choice to explore cough reflex and asthma [20], rabbits have, such as humans, undergone a significant distancing from the farm environment to a low microbial diversity environment that occurred over several generations. The specific housing procedures for rabbits (a pathogen-free zone for the control group and an experimental farm for the farm group) were implemented from one week of gestation to six weeks after birth to 'model' an early exposure followed by an exposure break.

## Material and methods

### Animals

Overall, six pregnant New Zealand rabbits were purchased from HYCOLE (SARL-HYCOLE) at one week of gestation. During housing, food (Safe® 110) and water were given ad libitum and checked daily by the technical staff. The enrichment consisted of hay or small pieces of wood. The animal care and study protocol was approved by the local ethics committee on animal testing (Comité d'éthique en expérimentation animale, CEEA) which is affiliated to the University of Lorraine (Comité d'Ethique Lorrain en Matière d'Experimentation animale CELMEA C2EA-66) followed by the validation of the Ministère de l'Enseignement Supérieur et de la Recherche under authorization number APAFIS#26171–2020051915444165 v3 according to recommendations 86–609 CEE issued by the council of the European Communities.

### Early life exposure

In order to study the early-life farm effect, pregnant rabbits were divided into two exposure groups according to the type of housing facility used. The rabbits and their offspring were kept in these specific facilities until six weeks after delivery.

The control group was housed in the pathogen-free zone of the Animal House of the University of Lorraine, in order to model an environment with low microbial diversity. In the pathogen-free zone, all material (including food and water) was sterilized before entering the zone. Animal handlers that entered the pathogen-free zone wore specific clothing that had previously been sterilized, as well as two pairs of gloves, overshoes, a mask, and a cap. The pregnant rabbits were housed in a wire mesh cage. Sterilized hemp litter was placed, without direct contact with the rabbits, under the cage to absorb excreta and was changed once a week. The newborn rabbits had a nest made of sterilized cardboard shavings. Regular sanitary controls performed during pregnancy and the offspring's first month of life revealed no pathogenic bacteria.

The farm group was housed in a calf barn at La ferme de la Bouzule, an experimental farm of the University of Lorraine. The calf barn comprises a small enclosed barn with limited wind and bird entry, where feed and bedding materials for calves are delivered by hand. The pregnant rabbits were housed in a wire mesh cage, directly next to the calves' pen, with hay and straw bedding that was changed once a week. The newborn rabbits had a nest made of straw with a direct entrance to their mothers' cage.

The rabbit feed was the same for both groups. From their seventh week of life, all the new-born rabbits were weaned and housed in the conventional zone of the Animal House of the University of Lorraine with a 16-hour day and 8-hour night cycle. After weaning, the rabbit mothers were proposed for adoption.

## Endotoxin levels in the farm environment

A sampling of airborne endotoxins was performed on three occasions, using a 37 mm polystyrene 3-piece closed-face cassette (Millipore®, France). The closed-face cassettes were mounted with a fiberglass filter (GF/B glass microfiber filter, pore size 1.0 μm, Whatman) as the collection support and another similar filter as the backing support. The fiberglass filters were previously heated at 250˚C for 120 minutes to make them pyrogen-free. The cassettes were connected to portable constant flow pumps (Apex2, Casella, USA) and the sampling was performed at a flow rate of 2 L.min$^{-1}$. The flow rate was calibrated and controlled before and after sampling using a bubble flow meter (Gilian®, Gilibrator, USA). All samples were taken at about 1.7 m from the ground.

The endotoxin concentration was assessed by introducing 5 mL of pyrogen-free water into each cassette, followed by shaking for 20 minutes at 2000 rpm (Multi-Reax® shaker, Heidolph®, Germany). The endotoxin concentration in the extract was assayed using the LAL kinetic chromogenic detection assay following the recommendations of the manufacturer, using Kinetic-QCL® kits (Lonza Group Ltd). The limit of detection of the endotoxin analysis was 0.005 EU.ml$^{-1}$, i.e., 0.025 EU per filter. The volume of air sampled was calculated based on the flow rate and the duration of sampling (around five hours), and the airborne endotoxin concentration was thus expressed in endotoxin units (EU) per cubic meter (EU.m$^{-3}$).

## Sensitization and provocation of airway inflammation with ovalbumin

The sensitization to ovalbumin (OVA) and challenge protocol used in this study are shown in Fig 1 and described in detail in the S1 File. The anti-OVA Immunoglobulin E (IgE) antibodies (undiluted sera) were determined by using the Rabbit Anti-OVA IgE ELISA kit from MyBioSource (San Diego, CA, USA), following the manufacturer's instructions.

## Anesthesia and animal preparation

The anesthesia, analgesia, euthanasia, and animal preparation are detailed in the S1 File. For technical reasons, we were forced to carry out the final experiments (anesthesia and provocation of cough) earlier (20 days before) in the farm group than in the control group.

## Mechanical provocation of cough in anesthetized and tracheotomized rabbits

The apparatus developed to elicit a discrete mechanical challenge to the trachea has been described in detail and validated in previous reports [21]. To describe it briefly, a rotating silastic catheter introduced in tracheotomy is driven by a small electric motor that spins the catheter and rubs its tip onto the airway mucosa for a short period. The electrical signal from the engines serves as a marker for accurate identification of the stimulus time course. As cough reflex is significantly more frequently provoked during inspiration compared to expiration, mechanical stimulations were triggered during the inspiratory phase. The beginning of inspiration was detected electronically as soon as the flow signal reached a positive value. Four stimulation durations (50, 150, 300, and 600 msec) that were each repeated twice in a pseudorandom order were performed. An interval of at least one minute of quiet and regular

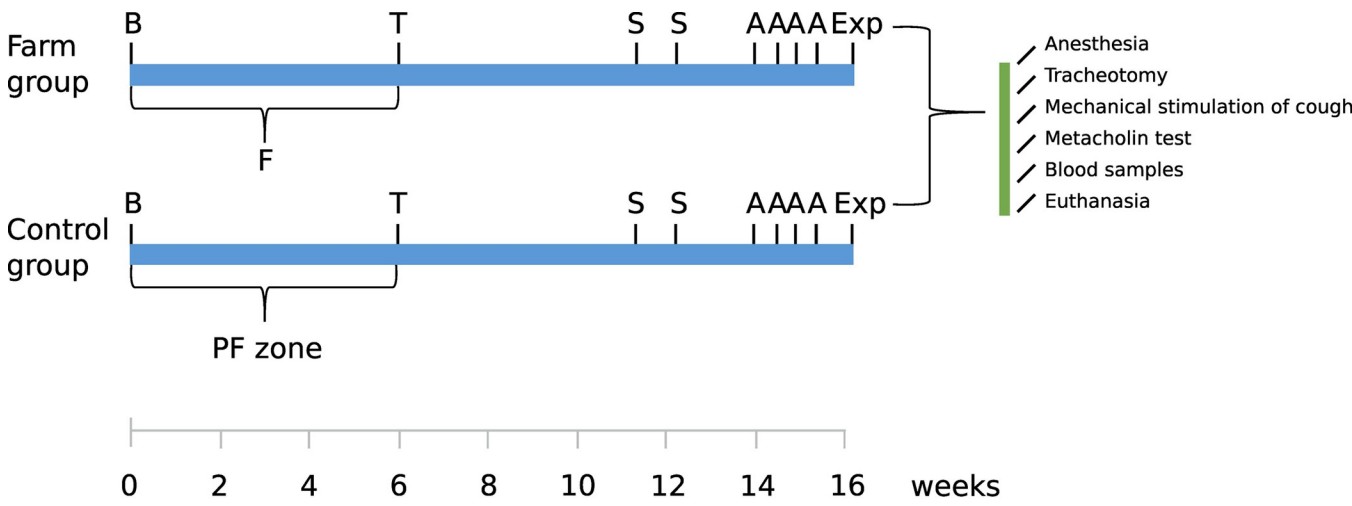

B: birth of rabbits
F: farm exposure
PF zone: pathogen-free zone exposure
T: transportation to the conventional zone of Animal House
S: sensitization to ovalbumin
A: aerosol with ovalbumin
Exp: final experimentation

**Fig 1. Sensitization and challenge protocol.**

breathing was allowed to elapse between two stimuli, during which the reference tidal volume (VT) was determined [22]. After each stimulation, four breath cycles were registered in order to study the type and intensity of the response.

The mechanical stimulation of the trachea provoked defensive reflexes (DR) that were further discriminated into a cough reflex (CR) and an expiration reflex (ER) according to the change in VT, peak expiratory flow (V'Epeak), and rectus abdominis electromyographic (EMG) activity [23]. The CR was defined by an increase in VT followed by an increase in V'Epeak associated with a burst of rectus abdominis EMG activity (Fig 2), and the ER was defined by an increase in V'Epeak without a preceding increase in VT associated with a burst of rectus abdominis EMG activity [24]. To take into account the spontaneous between-breath variability, an unbiased differentiation of the CR and the ER was achieved by a statistical evaluation of the VT between the stimulation and reference breaths. The VT of the reference breath was determined as the mean of three breaths prior to stimulation and its upper limit as the mean plus three standard deviations. The CR was identified when the VT of the stimulation breath was higher than the upper limit of the reference VT. The defensive response to one mechanical stimulation consisted of a bout of one or several CRs and/or ERs. In addition, three types of responses were considered while taking into account all the DRs induced by the mechanical stimulation during the next four breath cycles (not only the first response): an ER only, a CR only, and both an ER and a CR.

The use of four stimulation durations allowed the assessment of the duration–response curve and enabled the assessment of the defensive response threshold (DT) to mechanical stimulation. The DT was defined as the shortest stimulation duration necessary to provoke a

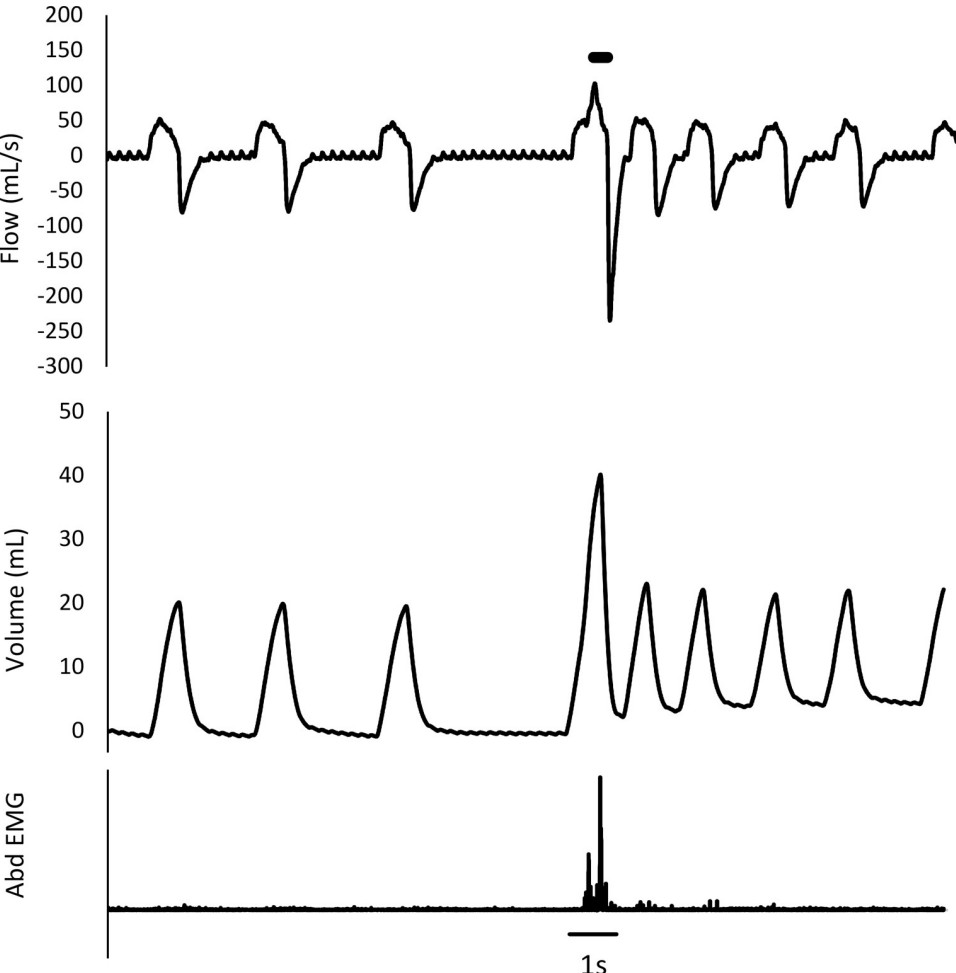

**Fig 2. Typical cough reflex to mechanical stimulation of the trachea.** The increased expiratory flow is associated with a burst of rectus abdominis EMG activity and preceded by an increase in the tidal volume. The thick bar on the top indicates the moment of mechanical stimulation.

response initiated by the CR (cough threshold [CT]) or ER (expiration reflex threshold [ET]). In the absence of any DR, even with the longest stimulation duration, the threshold was arbitrarily set to 1200 msec.

The maximal numbers of DR, CR, and ER were defined as the cumulative numbers of each reflex among the four cycles studied during one mechanical stimulation. The cumulative numbers of DR, CR, and ER were calculated as the sums of DR, CR, and ER numbers, respectively, induced by the eight mechanical stimulations performed on each rabbit.

## Measurement of allergen-induced airway responsiveness to methacholine

The respiratory measurements were performed as described previously [25]. The airflow was measured at the tracheal opening using a Fleisch pneumotachograph connected to a differential pressure transducer, the signal of which was integrated with the volume. The transrespiratory pressure was measured at a side port of the cannula using an identical pressure transducer. The airway caliber was assessed by the respiratory resistance (Rrs) using the forced oscillation technique. A loudspeaker generated a sine wave, forcing a signal at 20 Hz and the

Rrs was computed as the real part of the complex ratio of transrespiratory pressure to flow (respiratory impedance [Zrs]). The values near zero breathing flow were retained in order to minimize the flow-dependent component in Rrs. A methacholine (Mch) challenge was performed: successive quadrupled doses (0.016, 0.065, 0.25, 1, and 4mg.mL$^{-1}$) of Mch (Sigma-Aldrich, France) were delivered for 30 seconds by an ultrasonic nebulizer (LS 290, SYSTAM$^{®}$, France) connected to the tracheal cannula. Rrs was assessed after each Mch aerosol. Challenge was stopped when the resistance was superior or equal to a two-fold initial Rrs.

Two measurements of baseline Rrs during 60 seconds were performed and averaged thereafter to provide the baseline data for the Mch dose–response curves. Allergen-induced airway responsiveness to Mch was calculated as the Mch concentration needed to induce a 50% (PD50) and a 100% (PD100) increase in the Rrs when compared to baseline.

## Statistical analysis

All quantitative data were expressed by the median [IQR 25–75%], and statistical comparisons between groups were performed using non parametric Mann-Whitney U test. Statistical comparisons between groups for qualitative variables (DT, CT and ET) were performed using chi square test or exact fisher test. Correlation between quantitative variables was explored using Spearman's rank correlation coefficient. A non-parametric generalized linear mixed model was used to adjust the statistical comparison of quantitative variables with the correlated variables. Statistical analyzes were performed using SAS 9.4.

## Results

A total of six pregnant rabbits were used for this study with three rabbits and their offspring in each group. A total of 38 rabbit pups were included in the present study: 20 from the pathogen-free zone group, from now on called the control group (C), and 18 from the farm group (F).

At the time of final assessment of cough and bronchial reactivity, the rabbit pups in F were lighter (F: 2272 g [2075–2435] vs. C: 2637 g [2567–2797]; p<0.001) and younger (F: 90 days [84–96] vs. C: 110.5 days [104–117]; p<0.001) than the rabbit pups in C.

### Endotoxin level in the farm environment

Among the three samples of endotoxins in the calf barn, levels varied according to the days the samples were taken. The highest level was found in winter during the delivery of bedding material and the lowest in spring. The results were: 1854 EU.m$^{-3}$ in February, before the housing of pregnant rabbits, 109 EU.m$^{-3}$ in March, during the second week of pregnancy, and 30 EU.m$^{-3}$ at the beginning of May, when the rabbit pups were one month old.

### Sensitization to ovalbumin

Specific IgE to OVA assays were performed in 27 rabbits (16 from C and 11 from F). There was no difference in IgE sensitization to OVA between groups. The median was 4.66 μg.mL$^{-1}$ [4.07–6.19] in C vs 4.86 μg.mL$^{-1}$ [2.87–5.80] in F (p = 0.826).

### Cough and expiration reflexes

Results regarding chemical stimulation of cough in conscious rabbits are reported in S2 File.

Two rabbits in each group died at anesthesia induction. Thus, the following measurements are presented for 18 rabbits from C and 16 from F. There was no difference in defensive

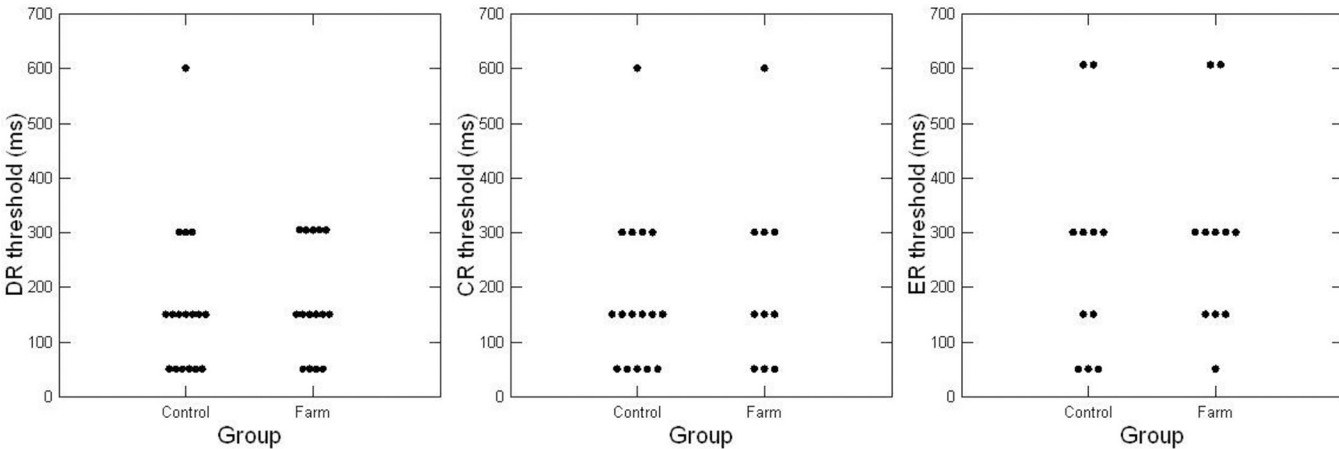

DR: defensive reflex; CR: cough reflex; ER: expiratory reflex

**Fig 3. Defensive reflexes threshold (in msec) in the control group and farm group.**

reflexes threshold between C and F (Fig 3, p = 0.862 for DT, p = 0.492 for CT, and p = 0.885 for ET). Types of DR did not differ between groups (Table 1, p = 0.249).

The estimated median of DT was 150 msec in both groups (Table 2). There was no difference between groups in the maximal numbers of DR, CR, and ER per stimulation or in the cumulative numbers of DR, CR, and ER induced by all the mechanical stimulations (Table 2).

There was no correlation in the intensity of DR, CR, and ER with age or weight (S1 Table). The intensity of ventilatory responses did not differ between groups (Table 3), even after adjustment for age and weight (Table 4).

## Bronchomotor responsiveness

Baseline Rrs was 23.13 hPa.s/L [21.17–27.43] in C and 21.86 hPa.s/L [18.06–25.66] in F and was not correlated with age or weight (S1 Table). There was no difference in baseline Rrs between groups whether before (Table 3) or after adjustment for weight (Table 4). Baseline Rrs was significantly higher in F after adjustment for age but this difference was no longer significant after adjustment for weight (p = 0.061, Table 4).

PD50 and PD100 were not correlated with baseline Rrs but there was a moderate correlation with weight and age (S1 Table). In the raw data analysis, there were significantly lower values for PD50 and PD100 in F (Table 3). However, after adjustment for age and/or weight, there was no significant difference between groups (Table 4).

## Discussion

In our study, early exposure of rabbits to farm environment did not significantly affect response to ovalbumin sensitization, ventilatory defensive reflexes provoked by mechanical

**Table 1. Type of defensive reflex.**

| Type of defensive response | Control Group | Farm Group |
|---|---|---|
| None | 1 | 1 |
| Expiratory only | 1 | 5 |
| Cough only | 6 | 4 |
| Mixed | 10 | 6 |

**Table 2. Defensive reflexes induced by mechanical stimulation (threshold, maximal number, and cumulative number).**

| | Control Group | Farm Group | p-value |
|---|---|---|---|
| | Median [Q1-Q3] | Median [Q1-Q3] | |
| **Defensive reflex threshold** | | | |
| Defensive Reflex | 150 [50–300] | 150 [100–300] | 0.862 |
| Cough Reflex | 150 [50–300] | 300 [150–1200] | 0.492 |
| Expiratory Reflex | 450 [150–1200] | 300 [225–1200] | 0.885 |
| **Maximal number of defensive reflexes per stimulation** | | | |
| Defensive Reflex | 2 [1–3] | 2.5 [1–3.5] | 0.448 |
| Cough Reflex | 1 [1–2] | 1 [0–2] | 0.270 |
| Expiratory Reflex | 1 [0–2] | 1 [1–2] | 0.342 |
| **Cumulative number of defensive reflexes** (induced by the eight mechanical stimulations) | | | |
| Defensive Reflex | 6 [4–8] | 6 [4–11] | 0.744 |
| Cough Reflex | 4 [2–6] | 2 [0.5–5.5] | 0.289 |
| Expiratory Reflex | 3 [0–3] | 2 [1–4.5] | 0.626 |

stimulations of the trachea, or bronchomotor responsiveness to methacholine. Endotoxin levels in the farm environment had great variability depending on the season and agricultural activities.

To the best of our knowledge, this is the first study designed to explore the role of early farm exposure in the development of cough and bronchial hypersensitivity in a rabbit model of allergic airway inflammation. Recent literature shows that the rabbit model is of particular interest for the study of asthma and other lung diseases for several reasons [20, 26], and we used this model especially because mice and rats, the most frequently used animals for asthma research, do not express a typical cough reflex (at least under anesthesia). Sensitivity to cough

**Table 3. Maximal intensity of ventilatory responses induced by mechanical stimulation and respiratory resistance before and after methacholine challenge.**

| | Control Group | Farm Group | p-value[*] |
|---|---|---|---|
| | Median [Q1-Q3] | Median [Q1-Q3] | |
| **Maximal intensity of ventilatory responses induced by mechanical stimulation** | | | |
| Maximal intensity of defensive reflex (%) | 269.24 [193.21–387.13] | 192.79 [144.46–349.40] | 0.132 |
| Maximal intensity of cough reflex (%) | 224.54 [137.00–406.12] | 227.29 [154.56–328.28] | 0.735 |
| Maximal intensity of expiratory reflex (%) | 191.33 [128.34–324.87] | 144.46 [123.60–229.07] | 0.368 |
| **Respiratory resistance before and after methacholine challenge** | | | |
| Baseline respiratory resistance (hPa.s/L) | 23.13 [21.17–27.43] | 21.86 [18.06–25.66] | 0.320 |
| PD50 (mg.mL$^{-1}$) | 0.30 [0.26–0.40] | 0.13 [0.07–0.19] | **0.026** |
| PD100 (mg.mL$^{-1}$) | 0.31 [0.26–0.75] | 0.12 [0.07–0.22] | **0.024** |

[*]Boldface values indicate $p < 0.05$.

**Table 4. Associations between exposures (farm vs control), intensity of respiratory responses induced by mechanical stimulation, and respiratory resistance before and after methacholine challenge.**

| | N | Model 1 | | Model 2 | | Model 3 | | Model 4 | |
|---|---|---|---|---|---|---|---|---|---|
| | | RR | 95% CI | RR | 95% CI | RR | 95% CI | RR | 95% CI |
| Maximal intensity of defensive reflex (%) | 32 | 3.06 | 0.62–15.09 | 5.59 | 0.47–66.30 | 2.66 | 0.22–31.82 | 4.86 | 0.15–152.09 |
| Maximal intensity of cough reflex (%) | 26 | 1.43 | 0.22–9.51 | 4.66 | 0.16–139.09 | 0.86 | 0.05–15.56 | 1.61 | 0.02–114.55 |
| Maximal intensity of expiratory reflex (%) | 22 | 2.13 | 0.42–10.77 | 2.42 | 0.12–48.99 | 1.56 | 0.23–10.35 | 2.60 | 0.08–82.67 |
| Baseline respiratory resistance (hPa.s/L) | 31 | 2.18 | 0.47–10.20 | **28.77** | **1.45–571.88** | 8.39 | 0.94–75.05 | 45.70 | 0.83->999.99 |
| PD50 (mg.mL$^{-1}$) | 30 | **6.61** | **1.41–31.00** | 0.78 | 0.07–8.34 | 4.93 | 0.69–35.18 | 0.72 | 0.05–9.33 |
| PD100 (mg.mL$^{-1}$) | 28 | **7.25** | **1.44–36.35** | 0.75 | 0.05–12.02 | 3.61 | 0.16–83.29 | 0.46 | 0.01–24.43 |

Boldface values are significant (p<0.05).

Model 1: Crude.

Model 2: Model 1 + age.

Model 3: Model 1 + weight.

Model 4: Model 1 + age + weight.

was explored by mechanical stimulation of trachea in anesthetized and tracheotomized rabbits. Whereas in human, cough can be elicited by inhalation of chemical agents (citric acid or capsaicin), rabbits are more responsive to mechanical stimulation than to chemical stimulation as shown in S2 File [27]. The nasal breathing of rabbits makes them poorly responsive to aerosol of citric acid and the scarce expression of TRPV1 receptors in the rabbit respiratory system makes them unresponsive to capsaicin [28, 29]. The response to OVA sensitization used to model allergic airway inflammation was assessed by specific IgE assays in a large proportion of rabbits. Our experimentation, with four OVA aerosols administered and the last one performed 48 hours before the final experiments, was designed to study antigen-induced bronchial hyperresponsiveness [26, 30–32]. Whereas most animal studies used farm dust samples as exposure, we applied farm environment in real life to pregnant rabbits and their offspring. Even though we characterized this farm environment by endotoxin measures, one limitation of this real-life approach is the variability of the farm environment, which is affected by several factors (season, ventilation, number of calves, mulch, etc.) [33]. Endotoxin levels in the pathogen free zone were not able to be tested because it was not possible to enter this zone with the sampling equipment. However, as the procedures to enter the zone and to care for the animals were intended to limit microbial exposure, we assume that endotoxin levels would have been very low.

Most animal and human studies focused on the protective effect of farm environment on asthma and atopic diseases without taking into account the associated cough hypersensitivity. In clinical settings, cough as a symptom is not specifically evaluated in the asthma control questionnaire [34] or test [35] and its evaluation is excluded from key asthma intervention trials. However, in the absence of infection, a cough provoked by different stimuli, and that worsens at night, is strongly associated with asthma. In our study, we chose to explore the impact of the farm environment on cough hyperreactivity because of the protective effect of farm on atopy and Th2 inflammation, one of the several etiological mechanisms of chronic cough hypersensitivity syndrome [36]. The absence of a significant protective effect of farm environment on cough hyperreactivity may be explained by other factors linked to the development of cough hyperreactivity and neuromodulation [37, 38]. Some studies suggest that dust and exposure to allergens may directly activate afferent nerves and provoke long-lasting cough hypersensitivity [39, 40]. However, up until now, no study has shown an increase in cough sensitivity after exposition to a farm environment. Whereas several human cohort studies have

shown a protective effect of farm environment on asthma [17, 41], our study failed to demonstrate the link between farm environment and bronchial responsiveness.

Previous animal studies using mice showed that chronic pre-exposure to lipopolysaccharide can reduce the sensitivity to house dust mite sensitization and suppress all of the key asthma features [42]. In our study, it seems that early exposure to farm did not affect sensitivity to OVA sensitization, bronchial responsiveness or cough sensitivity in response to OVA challenge. There may be several explanations for the absence of farm effect on atopic and respiratory outcomes in our study. First of all, farm exposure was not maintained during OVA-sensitization and challenge. Most studies have shown that even though early exposure is needed to obtain a protective effect [13], continuous exposure might also be needed. Furthermore, the route of sensitization influences the immune response. The intraperitoneal route is the route of choice in allergic rabbit models [20, 43]. It induces a stronger immune response than the intra-gastric, respiratory, or cutaneous route [44, 45] and is also less similar to real-life routes. Thus, early farm exposure might have been insufficient to protect against this strong route of sensitization. Further analysis of inflammatory mediators, cell distribution, and Th1/Th2 cytokines secretion in rabbits might help to better understand the impact of early exposure to farm dust on immune response to OVA sensitization.

Regarding the farm environment, the rabbits were housed in the calf barn where endotoxin levels were expected to be high due to the small enclosed area with limited wind entry and the use of hay and straw as feeding and bedding material. The study of allergy and asthma prevalence in Amish and Hutterites populations has shown that a high level of endotoxin exposure is needed to protect against allergy and asthma [19], and other studies have shown that there is a dose–response effect [3, 13]. However, the level of endotoxin exposure needed to protect against asthma has not been established yet. In our study, endotoxin levels found in the calf barn are similar to those described in other studies in cow sheds [46] and dairy cattle [47] from modern farms but lower than in older studies in dairy barns [48]. The wide range of endotoxin levels, with the highest one measured in winter after bedding activities, has already been described in cow sheds and dairy cattle or barns in a same place but with different settings [49]. Irrespective of the season during housing of pregnant rabbits and their offspring, the number of calves present in the barn might also affect the level of endotoxins [33].

Finally, not all farm environments protect against asthma and atopy [50]. In the past 20 years, traditional dairy farms have been studied in European cohorts to understand the specific factors of dairy farm environment involved in the protection of asthma and allergic diseases [5, 8, 9]. In addition to the exposure to a high endotoxin level, those studies have shown a protective effect of the consumption of raw cow's milk during pregnancy and in the first year of life [8] and also a protective effect of a diversity of early exposures [51], such as high food diversity [15] and exposure to different animal species [14]. Together with our results, these studies suggest that the protective effect of farm environment is not only due to the exposure to dairy farm's dust but also to several other factors. Finally, another interesting result of these cohorts is the variability of the protective effect of farm exposure according to genetic factors [14, 18, 52] that may be different in rabbits compared to humans.

## Conclusion

The advantage of animal models is the ability to analyze the impact of one factor at a time while controlling all other factors. In our study, we analyzed the relationship between early exposure to farm dust and allergic and respiratory outcomes in an allergic airway inflammation rabbit model. Early farm exposure during the rabbits' pregnancy and the offspring's six weeks of life did not affect allergic sensitization, cough sensitivity, and bronchial hyperactivity.

Unlike children living on a farm, in our animal model, the rabbits were exposed to a farm environment for a limited amount of time and were only exposed to farm dust. These results suggest that early exposure to farm dust alone is not sufficient and that continuous exposure to several sources of microbial diversity (dust, food, animals, etc.) is needed to prevent atopy and asthma.

## Supporting information

**S1 Table. Correlation between quantitative variables (Spearman's rank correlation coefficient).**
(DOCX)

**S2 Table. Defensive reflexes provoked by nebulization of acid citric aerosol.**
(DOCX)

**S1 File. Supporting methods.**
(DOCX)

**S2 File. Supporting results.**
(DOCX)

**S1 Dataset.**
(XLSX)

## Acknowledgments

We thank Alexandre Laflotte, director of the experimental farm, La Bouzule (ENSAIA), all the fieldworkers from the farm, all the fieldworkers from the animal house facility at the University of Lorraine; Lise Alonso and Xavier Simon from the National Institute of Research and Security (INRS) in Nancy for dust samples; and David Moulin from IMoPA at the University of Lorraine for his help with blood lab work.

## Author Contributions

**Conceptualization:** Amandine Divaret-Chauveau, Laurent Foucaud, Silvia Demoulin-Alexikova.

**Data curation:** Amandine Divaret-Chauveau, Bruno Demoulin, Silvia Demoulin-Alexikova.

**Formal analysis:** Amandine Divaret-Chauveau, Pauline Loison, Marcelo De Carvalho Bittencourt, Frédéric Mauny.

**Funding acquisition:** Amandine Divaret-Chauveau, Silvia Demoulin-Alexikova.

**Investigation:** Amandine Divaret-Chauveau, Laurent Foucaud, Bruno Demoulin, Cédric Teston, Pauline Loison, Silvia Demoulin-Alexikova.

**Methodology:** Laurent Foucaud, Bruno Demoulin, Cyril Schweitzer, Frédéric Mauny, Silvia Demoulin-Alexikova.

**Project administration:** Laurent Foucaud, Silvia Demoulin-Alexikova.

**Resources:** Laurent Foucaud, Bruno Demoulin, Cédric Teston, Pauline Loison, Pierre Le Cann, Marcelo De Carvalho Bittencourt, Silvia Demoulin-Alexikova.

**Supervision:** Laurent Foucaud, Frédéric Mauny, Silvia Demoulin-Alexikova.

**Validation:** Pierre Le Cann, Cyril Schweitzer, Marcelo De Carvalho Bittencourt.

**Writing – original draft:** Amandine Divaret-Chauveau.

**Writing – review & editing:** Amandine Divaret-Chauveau, Laurent Foucaud, Bruno Demoulin, Cédric Teston, Pauline Loison, Pierre Le Cann, Cyril Schweitzer, Marcelo De Carvalho Bittencourt, Frédéric Mauny, Silvia Demoulin-Alexikova.

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
