## [Decision Letter · Decision Letter 0]

1 Nov 2022

PONE-D-22-28448Early exposure to farm dust in an allergic airway inflammation rabbit model: Does it affect bronchial and cough hyperresponsiveness?PLOS ONE

Dear Dr. Amandine Divaret-Chauveau,

Thank you for submitting your manuscript to PLOS ONE. After careful consideration, we feel that it has merit but does not fully meet PLOS ONE’s publication criteria as it currently stands. Therefore, we invite you to submit a revised version of the manuscript that addresses the points raised during the review process.

We look forward to receiving your revised manuscript.

Kind regards,

Svetlana P. Chapoval

Academic Editor

PLOS ONE

Journal Requirements:

"A.D-C. reports support from ARAIRLOR for the present manuscript; a contract with the French Public Agency ANSES as an expert in allergy and pediatric; consulting fees for expertise in pediatric allergy for Stallergens, Aimmune Therapeutics and ALK; and support for attending meetings from Mead Johnson, Nutricia, Aimmune Therapeutics and Novartis Pharma SAS. F.M. reports a grant from the French public agency ANSES and participation on a Data Safety Monitoring Board for the clinical “Propila-Rifax”. All other authors have no conflict of interest in relation to this work."

Reviewers' comments:

Reviewer's Responses to Questions

**Comments to the Author**

1. Is the manuscript technically sound, and do the data support the conclusions?

Reviewer #1: Partly

2. Has the statistical analysis been performed appropriately and rigorously? 

Reviewer #1: Yes

3. Have the authors made all data underlying the findings in their manuscript fully available?

Reviewer #1: Yes

4. Is the manuscript presented in an intelligible fashion and written in standard English?

Reviewer #1: Yes

5. Review Comments to the Author

Reviewer #1: This is an interesting and generally well presented manuscript describing experiments trying to understand the potential protective role of a farm environment on allergen sensitisation and subsequent responses to allergen exposure. The latter have been measured as mechanically-induced cough reflexes and responses to methacholine as a measure of bronchial responsiveness. Whilst this is an topical area of research, I have a number of comments that require further explanation/clarification to improve the impact of the work:

1) From Figure 1 it appears that the two groups of rabbits were housed in different environments for six weeks, before both being transported a "controlled zone"in the animal unit of the University. They were then sensitised to ovalbumin several weeks later in this new environment. Why were the animals not sensitised to ovalbumin at the end of the initial exposure period whilst in situ in their two different environments? What was the rationale for leaving them several weeks before subsequently sensitising them in a new environment? This makes no sense whatsoever given what the authors claim they were trying to investigate.

2) The route of administration of the ovalbumin sensitisation was ip. Why was ovalbumin chosen as the allergen and not house dust mite as a more clinically relevant allergen? Also why was the ip route used for sensitisation rather than the more relevant nasal or inhalation route?

3) Rabbits are very sensitive to tussive agents such as inhaled citric acid or capsaicin that elicit cough in man. Why were responses to these agents not used rather than (or additional to) mechanical stimulation, not least because these could have looked at the cough sensitivity in conscious animals?

4) P17 - PD50 and PD100 were significantly lower for the F group compared with the C group when analysing the raw data, which disappeared when the results were adjusted for age/weight. However, on P13 of the manuscript the authors report that "pups in F were lighter and younger than rabbit pups in C". Surely these data mean that exposure to the farm environment has had some biological effect(s) and because of the differences at baseline the groups are not matched at the time of the cough and methacholne assessment meaning you should not have corrected for weight or age when assessing these effects in the two groups.

6. PLOS authors have the option to publish the peer review history of their article (what does this mean?). If published, this will include your full peer review and any attached files.

Reviewer #1: No

---

## [Author Response · Author response to Decision Letter 0]

5 Dec 2022

Review Comments to the Author

Reviewer #1: This is an interesting and generally well presented manuscript describing experiments trying to understand the potential protective role of a farm environment on allergen sensitisation and subsequent responses to allergen exposure. The latter have been measured as mechanically-induced cough reflexes and responses to methacholine as a measure of bronchial responsiveness. Whilst this is an topical area of research, I have a number of comments that require further explanation/clarification to improve the impact of the work:

1) From Figure 1 it appears that the two groups of rabbits were housed in different environments for six weeks, before both being transported a "controlled zone" in the animal unit of the University. They were then sensitised to ovalbumin several weeks later in this new environment. Why were the animals not sensitised to ovalbumin at the end of the initial exposure period whilst in situ in their two different environments? What was the rationale for leaving them several weeks before subsequently sensitising them in a new environment? This makes no sense whatsoever given what the authors claim they were trying to investigate.

Response: Indeed, rabbits were housed in the specific environments for six weeks after birth and then transported to the conventional zone of the animal unit of the University of Lorraine. Intraperitoneal sensitization was performed around their 12th week of life.

The design of the study was chosen in order to investigate if early farm exposure is sufficient to have a protective farm effect on asthma, even if sensitization happens later in life. 

Since few years, the first thousand days of life in human (from conception to the 2yrs old of the child) have been described as a window of opportunity for optimum health, growth and neurodevelopment. Regarding the immune system, this critical period of life seems to be a window of opportunity for immune homeostasis (1). 

Previous studies have shown that early farm exposure is associated with increased regulatory T-cell numbers and responsible for a switch in regulatory T-cells (2,3). Several studies, especially those from the PASTURE European cohort, have shown a protective effect of early farm exposure on the development of asthma, atopic dermatitis, rhinitis, sensitization and food allergies (4) but usually farm exposure is continuous at least until adulthood.

In order to provide adequate preventive measures, it is needed to know if early farm exposure followed by an exposure break is sufficient to provide prevention of allergic diseases, even when sensitization happens later in life.

We added two sentences in the introduction (lines 79-83 of the revised version of the manuscript) and changed the end of the introduction to explain why specific housing procedures were implemented only during pregnancy and six first weeks of life (lines 104- 107). 

2) The route of administration of the ovalbumin sensitisation was ip. Why was ovalbumin chosen as the allergen and not house dust mite as a more clinically relevant allergen? Also why was the ip route used for sensitisation rather than the more relevant nasal or inhalation route?

Response: Thank you for your accurate comment. It is true that nasal sensitization to house dust mite is a more physiological way of sensitization, that have gained interest in the last years. 

However, a few studies have used OVA or HDM sensitization in mice to induce allergic reactions without significant difference in the results (5,6). Nasal HDM sensitization is mostly used in mice with an abundance of literature using this model of allergic mice; in guinea pigs, we found only two studies using nasal HDM sensitization (7,8) and none in rabbits. As the volume needed for nasal sensitization depends on the size of the nasal cavity of the animal, one explanation can be that the volume of HDM needed for rabbits is too important and thus far too expensive.

Regarding routes of sensitization, even in the few studies that used aeroallergen sensitization in rabbits, rabbits were sensitized by intraperitoneal injections (9,10). We added a sentence in the discussion line 368 of the revised version of the manuscript to disclose that it’s the route of choice in rabbits.

In our study, we have chosen ip route to ovalbumin because it is the model of bronchial hyperreactivity in rabbits developed and experienced by the team in the University of Lorraine (11,12). The specific environmental exposure was already innovative, so we did not want to experience it in a new model for the team.

3) Rabbits are very sensitive to tussive agents such as inhaled citric acid or capsaicin that elicit cough in man. Why were responses to these agents not used rather than (or additional to) mechanical stimulation, not least because these could have looked at the cough sensitivity in conscious animals?

Response: Thank you for this interesting comments. Indeed, we have performed inhaled citric acid in conscious rabbits right before anaesthesia but we choose to remove it from the manuscript because only few rabbits were responders. As you asked for this information, it seems appropriate to put the results of chemical stimulation of cough in the manuscript, thus, we have added this information in the supporting information.

Only 3 and 4 rabbits of each group were responders to conscious chemical stimulation. We were not surprised by these results for two reasons: 

1/ Rabbits, as guinea pig, are known to have nasal breathing only. In consequence, in conscious animals, a huge majority of aerosol is deposed in nasal cavity and the deposit of citric acid on the trachea is really low compared to human who are stimulated with inhaled citric acid through a mouthpiece with mouth breathing only. 

2/ Furthermore, rabbits are described to be more mechano-sensitive than chemo-sensitive compared to guinea pigs (13,14). 

Finally, we did not use capsaicin because rabbit does not cough in response to capsaicin as TRPV1 receptors are scarcely expressed in the rabbit respiratory system (15). According to several publications, rabbits do not cough at all in response to capsaicin, but about 40% of them respond to citric acid with variable intensity (16,17). We have tried to perform an acid citric challenge in conscious rabbits using several citric acid concentrations in order to assess cough sensitivity. However, a great percentage of rabbits did not cough (84.2% in control and 77.8% in farm group). 

We added results regarding chemical stimulation in conscious rabbits in the supporting information and a paragraph in the discussion regarding the particularities of cough sensitivity in rabbits (lines 324-330 of the revised version of the manuscript).

4) P17 - PD50 and PD100 were significantly lower for the F group compared with the C group when analysing the raw data, which disappeared when the results were adjusted for age/weight. However, on P13 of the manuscript the authors report that "pups in F were lighter and younger than rabbit pups in C". Surely these data mean that exposure to the farm environment has had some biological effect(s) and because of the differences at baseline the groups are not matched at the time of the cough and methacholine assessment meaning you should not have corrected for weight or age when assessing these effects in the two groups.

Response: Thank you for pointing out this issue. Groups were not matched at the time of cough and methacholine assessment because we were forced to carry out the experiments a little earlier in the farm group for technical reasons (summer closure of the Animal House of the University of Lorraine). Thus, there was a difference of 20 days in the median age at the time of cough and methacholine assessment. Even if we cannot exclude that farming environment has an effect on the weight, most likely the difference in weight is due to the difference in age.

We have performed adjusted analyses on weight and age in order to explore if the difference in PD50 and PD100 was due to the baseline difference in age and weight or to the difference in environmental exposure.

We added a sentence in the methods (lines 176-178 of the revised version of the manuscript) to precise that final experiments were carried out earlier in the farm group. 

 

References

1. Torow N, Hornef MW. The Neonatal Window of Opportunity: Setting the Stage for Life-Long Host-Microbial Interaction and Immune Homeostasis. J Immunol. 2017 Jan 15;198(2):557–63. 

2. Lluis A, Depner M, Gaugler B, Saas P, Casaca VI, Raedler D, et al. Increased regulatory T-cell numbers are associated with farm milk exposure and lower atopic sensitization and asthma in childhood. J Allergy Clin Immunol. 2014 Feb;133(2):551–9. 

3. Schröder PC, Illi S, Casaca VI, Lluis A, Böck A, Roduit C, et al. A switch in regulatory T cells through farm exposure during immune maturation in childhood. Allergy. 2017 Apr;72(4):604–15. 

4. Vuitton DA, Divaret-Chauveau A, Dalphin ML, Laplante JJ, von Mutius E, Dalphin JC. Protection contre l’allergie par l’environnement de la ferme : en 15 ans, qu’avons-nous appris de la cohorte européenne « PASTURE » ? Bull Académie Natl Médecine. 2019 Oct 1;203(7):618–30.

5. Nowroozilarki N, Öz HH, Schroth C, Hector A, Nürnberg B, Hartl D, et al. Anti-inflammatory role of CD11b+Ly6G+ neutrophilic cells in allergic airway inflammation in mice. Immunol Lett. 2018 Dec;204:67–74. 

6. Ding FX, Liu B, Zou WJ, Li QB, Tian DY, Fu Z. Pseudomonas aeruginosa-derived exosomes ameliorates allergic reactions via inducing the Treg response in asthma. Pediatr Res. 2018 Jul;84(1):125–33. 

7. Buday T, Gavliakova S, Mokry J, Medvedova I, Kavalcikova-Bogdanova N, Plevkova J. The Guinea Pig Sensitized by House Dust Mite: A Model of Experimental Cough Studies. Adv Exp Med Biol. 2016;905:87–95. 

8. Ramos-Ramírez P, Noreby M, Liu J, Ji J, Abdillahi SM, Olsson H, et al. A new house dust mite-driven and mast cell-activated model of asthma in the guinea pig. Clin Exp Allergy. 2020 Oct;50(10):1184–95. 

9. Patel HJ, Douglas GJ, Herd CM, Spina D, Giembycz MA, Barnes PJ, et al. Antigen-induced bronchial hyperresponsiveness in the rabbit is not dependent on M(2)-receptor dysfunction. Pulm Pharmacol Ther. 1999;12(4):245–55. 

10. Keir SD, Spina D, Douglas G, Herd C, Page CP. Airway responsiveness in an allergic rabbit model. J Pharmacol Toxicol Methods. 2011 Oct;64(2):187–95. 

11. Tiotiu A, Chenuel B, Foucaud L, Demoulin B, Demoulin-Alexikova S, Christov C, et al. Lack of desensitization of the cough reflex in ovalbumin-sensitized rabbits during exercise. PLoS ONE. 2017;12(2):e0171862. 

12. Foucaud L, Demoulin B, Leblanc AL, Ioan I, Schweitzer C, Demoulin-Alexikova S. Modulation of protective reflex cough by acute immune driven inflammation of lower airways in anesthetized rabbits. PLoS One. 2019;14(12):e0226442. 

13. Tatár M, Pécová R, Karcolová D. [Sensitivity of the cough reflex in awake guinea pigs, rats and rabbits]. Bratisl Lek Listy. 1997 Oct;98(10):539–43. 

14. Hanácek J, Davies A, Widdicombe JG. Influence of lung stretch receptors on the cough reflex in rabbits. Respiration. 1984;45(3):161–8. 

15. Tatar M, Hanacek J, Widdicombe J. The expiration reflex from the trachea and bronchi. Eur Respir J. 2008 Feb;31(2):385–90. 

16. Adcock JJ, Douglas GJ, Garabette M, Gascoigne M, Beatch G, Walker M, et al. RSD931, a novel anti-tussive agent acting on airway sensory nerves. Br J Pharmacol. 2003 Feb;138(3):407–16. 

17. Mutolo D, Cinelli E, Iovino L, Pantaleo T, Bongianni F. Downregulation of the cough reflex by aclidinium and tiotropium in awake and anesthetized rabbits. Pulm Pharmacol Ther. 2016 Jun;38:1–9.

---

## [Decision Letter · Decision Letter 1]

8 Dec 2022

Early exposure to farm dust in an allergic airway inflammation rabbit model: Does it affect bronchial and cough hyperresponsiveness?

PONE-D-22-28448R1

Dear Dr. Amandine Divaret-Chauveau,

We’re pleased to inform you that your manuscript has been judged scientifically suitable for publication and will be formally accepted for publication once it meets all outstanding technical requirements.

Kind regards,

Svetlana P. Chapoval

Academic Editor

PLOS ONE

Reviewers' comments:

Reviewer's Responses to Questions

**Comments to the Author**

1. If the authors have adequately addressed your comments raised in a previous round of review and you feel that this manuscript is now acceptable for publication, you may indicate that here to bypass the “Comments to the Author” section, enter your conflict of interest statement in the “Confidential to Editor” section, and submit your "Accept" recommendation.

Reviewer #1: All comments have been addressed

2. Is the manuscript technically sound, and do the data support the conclusions?

Reviewer #1: Yes

3. Has the statistical analysis been performed appropriately and rigorously? 

Reviewer #1: Yes

4. Have the authors made all data underlying the findings in their manuscript fully available?

Reviewer #1: Yes

5. Is the manuscript presented in an intelligible fashion and written in standard English?

Reviewer #1: Yes

6. Review Comments to the Author

Reviewer #1: The revised version has addressed the issues raised adequately. My only additional comment would be to make better reference to the work with inhaled citric acid in the main text, rather than just in the Supplementary Material.

7. PLOS authors have the option to publish the peer review history of their article (what does this mean?). If published, this will include your full peer review and any attached files.

Reviewer #1: No

---

## [Editor Report · Acceptance letter]

18 Jan 2023

PONE-D-22-28448R1 

Early exposure to farm dust in an allergic airway inflammation rabbit model: Does it affect bronchial and cough hyperresponsiveness? 

Dear Dr. Divaret-Chauveau:

I'm pleased to inform you that your manuscript has been deemed suitable for publication in PLOS ONE. Congratulations! Your manuscript is now with our production department. 

Kind regards, 

on behalf of

Dr. Svetlana P. Chapoval 

Academic Editor

PLOS ONE